# Learning partially observable PDE dynamics with neural networks

## Abstract

Spatio-Temporal processes bear a central importance in many applied scientific fields. Generally, differential equations are used to describe these processes. In this work, we address the problem of learning spatio-temporal dynamics with neural networks when only partial information on the system's state is available. Taking inspiration from the dynamical system approach, we outline a general framework in which complex dynamics generated by families of differential equations can be learned in a principled way. Two models are derived from this framework. We demonstrate how they can be applied in practice by considering the problem of forecasting fluid flows. We show how the underlying equations fit into our formalism and evaluate our method by comparing with standard baselines.

## 1 Introduction

We consider the problem of learning complex dynamics typical of physical systems using deep learning models. Modeling these dynamics is central to almost any problem in applied physics. The classical approach to this problem is model based : the dynamics of the physical system is usually formulated using Partial Differential Equations (PDEs) describing the evolution of state variables. PDEs are then solved through some clever numerical scheme (as in Meerschaert & Tadjeran (2006) for example). This methodology is used in many application domains like climate science, financial engineering, etc.. Although it has been extensively developed, several challenges remain open. Modeling physical phenomena accurately can be tedious, if not unfeasible : the relations between state variables are often complex and, even when those are known, specifying a functional form for the dynamics can be difficult (Lions et al. (1992), Peebles (1980)). Moreover, designing accurate, robust and computationally feasible numerical schemes for PDEs is usually a difficult problem for which current solutions rely on extensive domain specific background.

The availability of huge data quantities gathered from different types of sensors or simulations is a strong incentive to develop data driven approaches, as an alternative or complement to standard model based methods. For many real world problems, the underlying physics of the phenomenon is not always known in its entirety and data-driven methods, inspired by our current physical knowledge, appear as a particularly promising alternative to model based approaches, see Bezenac et al. (2018), Ling et al. (2016), Stewart & Ermon (2016) or Rudy et al. (2017b) for some examples. On the other hand, mature fields such as differential equations could provide machine learning with useful insights and intuitions. We advocate here a ML approach incorporating structural priors inspired from such a well developed domain.

We consider the problem of learning dynamical systems where available observations only provide partial information on the system state and dynamics which is a typical scenario in real world applications, be it in weather forecasting, financial engineering, or even video prediction. The objective is to learn from such observations the underlying system dynamics so as to forecast future observations. We focus on data generated by complex highly non-linear differential equations and analyze whether these dynamics can be learned by Deep Learning models. This context is particularly interesting since simulations relying on PDEs are essential in many applied disciplines and since this allows us to perform controlled experiments providing access to all the variables of the problem.

Our contributions are the following :

- We propose a general framework for modeling partially observed dynamical systems governed by PDEs with neural networks. This framework is inspired by the way physicists incorporate observations into their forecasting pipeline. It should be flexible enough to accommodate a large class of modeling problems other than forecasting observations.

- From this framework we derive two instances of training (PT and JT), with two different levels of prior injection, and two methods for multi-step forecasting (SSE and MSRE).

- As a case study, we consider two PDEs describing the evolution of fluid flow, namely the *Euler* and *Navier-Stokes equations*, and show how they can be formulated within this formalism. We study the performance and behaviour of our models on the problem of forecasting fluid flow using two common baseline models and focusing on the interplay between injection of prior information and performance.

We introduce our framework in section 2, detail the framework and its instances in section 3 and show how PDEs fit within this framework in section 4. Experimental results and analysis are described in section 5 and recent related work is reviewed in section 6.

## 2 BACKGROUND AND PROBLEM SETTING

### 2.1 LEARNING FROM PARTIALLY OBSERVED DATA

A dynamical system in our context can be broadly defined as a function $X$ with values in $\Omega$ which obeys a set of differential equations. We consider spatio-temporal dynamics for which $X$ can be written as a function of $(t, x) \in \mathbb{R} \times \mathbb{R}^d$ where $t$ and $x$ are respectively the time and space variables. The spatial vector-valued function $X_t$ contains the quantities of interest describing a studied physical system at time $t$ and gives the evolution in time and space of the system's state. More precisely, we will be interested in differential equations which can be written as :

$$\frac{dX_t}{dt} = F(X_t, \mathcal{D}_x X_t) \tag{1}$$

where $X_t$ denotes the state vector at time $t$, $\mathcal{D}_x X_t$ the spatial derivatives of $X_t$ and $F$ is a function with values in $\Omega$ (common instances for $F$ involve the gradient operator $\nabla X$, the Laplacian $\nabla^2 X$,...). By specifying a particular structure on $F$, one can describe the dynamics of many different classes of systems : an important example is that of the Navier-Stokes equations from fluid dynamics which bear a central importance in many applied sciences and which will be our application example.

### 2.1.1 NEURAL IMPLEMENTATION OF NUMERICAL SCHEMES

Solving differential equations usually requires using a numerical scheme in order to approximate the solution. Finite difference methods for example make use of discrete approximations of the derivatives. Let us suppose available a sequence of states $\{(X_t)_t\}$, each $X_t$ being a function of the space variable $x$, $X_t$ and $X_{t+\delta t}$ being separated by a short time interval $\delta t$ for all $t$. A straightforward discretization of equation 1 is the forward Euler scheme :

$$X_{t+\delta t} - X_t \approx \delta t \cdot F(X_t, \mathcal{D}_x X_t)$$

where $\delta t$ is the discretization timestep.

Using a parameterized neural network $r_\theta$, we can train it in order to fit the relation above :

$$X_{t+\delta t} \approx X_t + \delta t \cdot r_\theta(X_t, \mathcal{D}_x X_t)$$

This is reminiscent of the skip connection structures used in the residual block of ResNet or in some RNN models. Linear multi-step methods extend this idea by performing several integration steps. A simple multi-step extension of the Euler scheme could then be :

$$X_{t+k\delta t} \approx X_t + \delta t \sum_{l=0}^{k-1} r_\theta(X_{t+l\delta t}, \mathcal{D}_x X_{t+l\delta t})$$

This sequential process could be implemented by a ResNet architecture so that the latter could be used as a parametric model for approximating solutions of differential equations.

Moreover, spatial derivatives of $X_t$ at a given point $x$ can classically be expressed with finite difference schemes over the values of the function at neighbouring points $V_x$ :

$$\mathcal{D}X_t(x) = K(\{X_t(x') \mid x' \in V_x\})$$

where $K$ is a linear function. Thus, we can express each component of $\mathcal{D}_x X_t$ as a convolution over $X_t$, motivating the use of convolutional networks. Thus, given a sufficiently large training set of state sequences and supposing that we are able to learn their dynamics, we could, just by knowing the initial state of our system, use the learned operator and forecast its behaviour for any time horizon.

One could constrain the structure of $K$ to express a particular set of derivatives. This would be a way to add additional structure and knowledge into the forecasting operator. One could also adopt a more hybrid approach such as in Long et al. (2018). For the examples we studied in this paper, we use standard convolutional ResNets as we wanted to disentangle as much as possible the contributions of the elements we introduce, particularly when injecting priors into the estimator. However, it would be interesting in future research to analyze the effects of more structural constraints added into the forecasting operator.

### 2.1.2 Linking states to partial observations

In realistic settings, the state is generally only partially observed *e.g.*, when studying the ocean's circulation, variables contained in the system's state such as temperature or salinity are observable while others such as velocity or pressure are not. In other words, the measured data is only a projection of the complete state $X$. We model this measurement process with a fixed operator $\mathcal{H}$ linking the system's state $X_t$ to the corresponding observation $Y_t$ :

$$Y_t = \mathcal{H}(X_t)$$

In the following, $\mathcal{H}$ is supposed to be known, fixed and differentiable. This setting is the one which we focus on in this paper.

## 3 Framework

To summarize, observations $Y_t$ are supposed to be generated by a state space model with variable $X_t$ and the main objective is the prediction of future observations $Y_{t+k}$ for a certain horizon k. The general form of the model for a prediction at horizon 1 is then :

$$\begin{cases} \text{initial state } X_0 \\ X_{t+1} = g(X_t) \\ Y_t = \mathcal{H}(X_t) \end{cases} \tag{2}$$

We make the simplifying hypothesis that the information loss through the projection $\mathcal{H}$ is so that a state $X_t$ can be deterministically reconstructed using a long enough sequence. In other words, there exists an integer $k$ and an operator $\mathcal{E}$ such that :

$$X_t = \mathcal{E}(Y_{t-k+1}, ..., Y_t) \tag{3}$$

For more details on this assumption, refer to appendix C.

For now, let us consider only one step predictions : given a sequence of initial observations $(Y_1, ..., Y_k)$, we want to find the best estimation $\widehat{Y}_{k+1}$ for the observation $Y_{k+1}$, where observations $Y$ are supposed to be generated an underlying process $X$. We propose a framework where predicting $\widehat{Y}_{k+1}$ is decomposed into three steps :

- Prediction of the current state : Given past observations, we use a learnable operator $e_\omega$ parameterized by $\omega$ to produce a (complete) state vector $\widetilde{X}_t$.

- Prediction of the future state : $\widetilde{X}_t$ is given as input to a second learnable operator $f_\theta$ parameterized by $\theta$ which produces the next state $\widehat{X}_{t+1}$.

- Projection of the future observation : $\widehat{X}_{t+1}$ is mapped onto the space of observations using operator $\mathcal{H}$ to predict future observation $\widehat{Y}_{t+1}$.

This is summarized in the following system :

$$\begin{cases} \widetilde{X}_t = e_\omega(Y_{t-k+1}, ..., Y_t) \\ \qquad \widehat{X}_{t+1} = f_\theta(\widetilde{X}_t) \\ \qquad \widehat{Y}_{t+1} = \mathcal{H}(\widetilde{X}_{t+1}) \end{cases} \tag{4}$$

The learning problems amounts to optimizing the following loss :

$$\min_{\theta,\omega} \mathbb{E}_{(Y_1,...,Y_{k+1})\in\text{Data}}[d(\mathcal{H}(f_\theta(e_\omega(Y_1, ..., Y_k)))), Y_{k+1})] \tag{5}$$

where $d$ is the loss metric and $e$ and $f$ are functions belonging to appropriate parametric families. Time lag parameter $k$ can be set up by cross-validation. As discussed in section 2, a natural choice for $f$ is a convolutional residual network modeling the system's dynamics whereas, for $e$, designing the right family of functions will depend on the structure of the studied data.

## 3.1 FORECASTING MODELS

Let us now introduce two instances of the framework described above. For clarity, we first consider one step predictions and then introduce the multi-step extension in section 3.2.

### 3.1.1 JOINT TRAINING OF ESTIMATION AND FORECASTING

When the information about the underlying process is limited to observations, the straightforward way to solve the minimization problem equation 5 is to train $e$ and $f$ jointly. Prior information here is limited to the architectures of $e$ and $f$ as well as that of the pipeline equation 4 and the fixed operator $\mathcal{H}$. Let $\left\{ \left( (Y_{t-k+1}^{(i)}, ..., Y_t^{(i)}), Y_{t+1}^{(i)} \right) \right\}_i$ be a training set of observation sequences, each sequence being identified by index $i$. The loss corresponding to sequence $i$ is then :

$$\mathcal{L} = d\big[\mathcal{H}(f_\theta(e_\omega(Y_{t-k+1}^{(i)}, ..., Y_t^{(i)}))), Y_{t+1}^{(i)}\big]$$

With this approach only observations are needed for training without any additional supervision. On the other hand, there is no reason for it to have any interpretable structure or correspond to the physical modelling of the studied system.

### 3.1.2 PRE-TRAINING THE STATE ESTIMATOR

It is not uncommon that some form of additional knowledge on the system state is available and can be exploited.

- It is sometimes possible to access to a limited number of examples of state sequences through costly measurements. Then pre-training of $e$ and $f$ could be beneficial, its success depending on the complexity of the studied dynamics and on the information loss associated with $\mathcal{H}$.
- More often it is possible to have access to a noisy or approximate version of the states $X_t$ through simulations. Complex simulators and generated data are available in several domains : it is common for models to be developed using simulated data and then to be adapted to real observations. A lighter alternative is to use a simplified model which allows fast and cheap simulations. Generated sequences can then be used e.g. for pre-training $e$ and $f$ before fine-tuning with real observations or more complex simulations. In both cases, the simplified dynamic plays the role of a prior over the underlying process. In order to illustrate and evaluate this idea, we will consider the Euler equations as a simplified version of the more complex Navier Stokes equations in the experiments (section 5.3).

## 3.2 MULTI-STEP FORECASTING

We present below two strategies that could be used for prediction at horizon $l > 1$. In order to forecast $l \geqslant 2$ steps ahead, once the state at time $t$ has been the estimated with $\widetilde{X}_t$ using $e_\omega$, we may apply one of the following strategies :

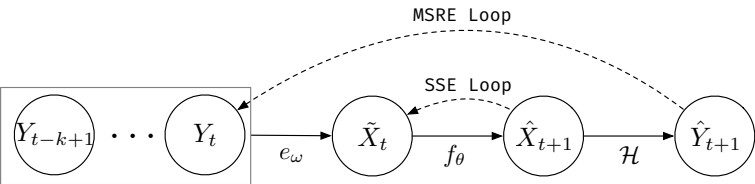

Figure 1: Illustration of the two multi-steps prediction strategies - SSE and MSRE - introduced in section 3.2.

- Single State Estimation (**SSE**) : the estimation operator $e$ is used only once in order to obtain an initial state $\widetilde{X}_t$ for the forecasting operator:

$$\widehat{X}_{t+l} = (f_\theta \circ \cdots \circ f_\theta)(\widetilde{X}_t)$$

- Multiple State REstimation (**MSRE**) : We use the forecast operator to produce the next state

$$\widehat{X}_{t+1} = f_\theta(\widetilde{X}_t)$$

Then, for the forecast at $t+2$, we estimate the state at time $t+1$ by plugging the generated observation $\mathcal{H}(\widehat{X}_{t+1})$ in the estimation operator $e_\omega$ :

$$\widetilde{X}_{t+1} = e_\omega(Y_{t-k+2}, ..., Y_t, \mathcal{H}(\widehat{X}_{t+1}))$$

so that:

$$\widehat{X}_{t+2} = f_\theta(\widetilde{X}_{t+1})$$

and then iterate for the following steps.

In both cases, ground truth observations up to $Y_{t+l}$ will be used as targets in the loss term when learning to predict at horizon $l$.

The SSE method is the classical one in dynamical systems. However, especially when forecasting for long time horizons, it supposes that the estimator $e$ is able to compress all relevant information about the long term evolution of the system into the initial state, which is a difficult task. On the other hand, the MSRE strategy, while less natural from the physical point of view, allows to gather information from the observations space at each step which should be an easier task.

Note that using a multi-step forecasting objective during training could be useful since it imposes additional constraints on the learned functions. This will be explored in the experimental section. When using multi-step forecasting at training time, the error is back-propagated through the $l$ prediction steps and one has to choose a principled way of supervising the training. Moreover, we compare in appendix E different sampling strategies for long sequences training. Finally, following the analysis done in Saxe et al. (2013), we have observed that orthogonal initialization of the weights of $f$ is also important.

## 4 EXAMPLES FROM FLUID DYNAMICS : EULER AND NAVIER-STOKES

As an illustration, we now focus on the problem of predicting the evolution of a two-dimensional fluid mechanics system. The equations governing fluids are among the most studied. One of the challenging aspects of those highly non-linear equations is the chaotic behaviour of turbulent flows. We use, as examples, instances of Euler and Navier-Stokes equations which are briefly introduced below. More information is available in appendix A, and Acheson (1989) is a comprehensive reference textbook. Euler being a simplified version of Navier Stokes, only the latter is presented below.

Navier Stokes equations for incompressible fluids are :

$$\begin{cases} \dfrac{\partial u}{\partial t} + (u \cdot \nabla)u = -\dfrac{\nabla p}{\rho} + g + \nu \nabla^2 u \\ \qquad\qquad \dfrac{\partial \rho}{\partial t} + (u \cdot \nabla)\rho = 0 \\ \qquad\qquad\qquad \nabla \cdot u = 0 \end{cases} \tag{6}$$

The first equation results from Newton's second law, the second comes from the conservation of density while the third is the incompressibility condition. In those equations, $u$ denotes a two-dimensional velocity field, $p$ is a pressure scalar field, $\rho$ is the fluid density and $g$ is the gravitational force. $\nabla \cdot u$ is the divergence of velocity $u$, $u \cdot \nabla$ is the advection operator $u \cdot \nabla = u_x \dfrac{\partial.}{\partial x} + u_y \dfrac{\partial.}{\partial y}$

Euler equations are similar to Navier Stokes but do not consider the particular nature of the fluid and ignore the viscosity term $\nu \nabla^2 u$ in the first equation of (6) by taking $\nu = 0$.

For Navier-Stokes, $\nu$ is the *kinematic viscosity* and $\nabla^2$ is the Lagrangian operator. The importance of viscosity in a fluid, and thus a way to assess its qualitative behaviour, cannot be measured through $\nu$ alone. The most commonly used quantity to describe it is the *Reynolds number $R$*. This is the ratio of the inertia and the viscosity terms. If $R$ is small, it means that the fluid is very viscous, behaving more like honey ; If $R$ is large, the viscosity term is small and the fluid behaves more like water. A definition of the *Reynolds number* is provided in appendix A.

Equation equation 6 is not of the same form as equation 1 as we still have the pressure variable $p$ as well as the null divergence constraint. It is, however, possible to show that it follows the general form of equation 1 as there exists an operator $\mathbb{P}$ which outputs divergence-free vector fields so that, for Navier Stokes, the third equation is dropped and the first equation takes the following form :

$$\frac{\partial u}{\partial t} = \mathbb{P}[\int -(u \cdot \nabla)u + \nu \nabla^2 u]$$

This is similar to equation 1 so that we can put the whole system in that form with $X = (\rho, u)$. This means that all the results and developments in previous sections apply for this family of fluid equations. We give more details about this in the appendix B[1]

Note that in what follows, we have $\mathcal{H}(X) = \rho$. In other words, we observe the scalar field of densities and the complete state is obtained by concatenating the 2D velocity flow $u$ to $\rho$.

## 5 EXPERIMENTS

### 5.1 DETAILS

The architecture of $e_\omega$ is based on the UNet presented in Ronneberger et al. (2015) which is a convolutional-deconvolutional network with skip connections which takes as input $k$ feature maps corresponding to the $k$ different observations. In all the following experiments, we have chosen $k = 3$. For the forecasting operator $f_\theta$, we use the ResNet architecture He et al. (2016) with Leaky ReLU non-linearities of parameter 0.1 and 6 residual blocks.

Our data is generated from a numerical simulation, using the Mantaflow fluid simulation library Thuerey & Pfaff (2018). We study three different dynamics : the first one is governed by the Euler equations, the second and the third by the Navier-Stokes equations with $R = 5000$ and $R = 100000$. The former corresponds to a viscous fluid with complex dynamics while the latter corresponds to a low viscosity, closer to Euler.

For each setting, we produce 300 training sequences of 500 timesteps each (one timestep being of duration $0.5s$), randomly selecting initial conditions for each sequence. We then subsample 5 times in time (two successive states are then separated by $2.5s$). In order to tune the hyperparameters of our experiments, we generate a validation set of 200 as well as a test set of 100 sequences generated randomly and independently.

As a normalization strategy, we chose dividing each component of the state by its empirical standard deviation over the dataset. We do not set the mean to zero as the positivity of the density is an important constraint of the problem. All training horizons are set to $l = 8$. A more thorough exploration of this parameter is done in appendix D.

As for the baselines, we have used (1) a ResNet which takes $k = 3$ past observations as input to produce a future observation directly (for multi-step forecasting, we just give the produced obser-

---

[1]Let us note that, while this might just seem like a mere technicality, it still provides us with an important information : forecasting the dynamics of those equations while following equation 1 automatically makes us find a null-divergence velocity vector field.

vations back to the ResNet in an auto-regressive manner), which can be seen as an instance of our framework with $e$ being the identity operator, and (2) a Convolutional LSTM (Shi et al. (2015)) with 2 layers (we have tried LSTMs with different numbers of layers and hidden dimensions and selected the one with the lowest validation error).

## 5.2 FORECASTING PERFORMANCE FOR EULER AND NAVIER-STOKES EQUATIONS

|  | $t_0 + 1$ | $t_0 + 5$ | $t_0 + 10$ | $t_0 + 15$ | $t_0 + 20$ | $t_0 + 25$ | $t_0 + 30$ |
|---|---|---|---|---|---|---|---|
| **ResNet** | 0.042 | 0.106 | 0.195 | 0.267 | 0.324 | 0.370 | 0.407 |
| **ConvLSTM** | 0.16 | 0.24 | 0.35 | 0.47 | 0.57 | 0.70 | 0.90 |
| **PT MSRE** | 0.010 | 0.071 | 0.102 | 0.257 | 0.324 | 0.377 | 0.419 |
| **PT SSE** | 0.011 | 0.059 | 0.102 | 0.179 | **0.234** | 0.288 | 0.341 |
| **JT MSRE** | **0.007** | **0.044** | **0.10** | **0.176** | 0.235 | **0.284** | **0.327** |
| **JT SSE** | 0.308 | 0.599 | 0.594 | 1.27 | 5.24 | 15.4 | 33.5 |

Table 1: Euler equations: test average MSE for two baselines (ResNet and ConvLSTM) and 4 instances of the proposed model for different forecasting horizons. PT and JT are respectively the pre-trained and jointly trained models, SSE and MSRE the two multi-step training strategies introduced in section 3.2.

|  | $t_0 + 1$ | $t_0 + 5$ | $t_0 + 10$ | $t_0 + 15$ | $t_0 + 20$ | $t_0 + 25$ | $t_0 + 30$ |
|---|---|---|---|---|---|---|---|
| **Jointly trained** | **0.00097** | **0.0066** | **0.0196** | **0.0376** | **0.0588** | **0.0824** | **0.107** |
| **Pre-trained estimator** | 0.017 | 0.108 | 0.273 | 0.373 | 0.446 | 0.512 | 0.579 |
| **Fine-tuned Pre-Trained estimator** | 0.0016 | 0.0105 | 0.0256 | 0.0455 | 0.0742 | 0.105 | 0.169 |
| **ResNet** | 0.0070 | 0.0181 | 0.0423 | 0.0720 | 0.104 | 0.137 | 0.171 |

Table 2: Navier-Stokes equations with $R = 5000$ : Test average MSE per time-step for a Jointly Trained system, a system with pre-trained estimator and one with a pre-trained estimator fine-tuned on the Navier-Stokes data, for different forecasting horizons.

|  | $t_0 + 1$ | $t_0 + 5$ | $t_0 + 10$ | $t_0 + 15$ | $t_0 + 20$ | $t_0 + 25$ | $t_0 + 30$ |
|---|---|---|---|---|---|---|---|
| **Jointly trained** | **0.0091** | **0.047** | 0.104 | 0.18 | 0.24 | **0.288** | 0.333 |
| **Fine-tuned Pre-Trained estimator** | 0.011 | 0.049 | **0.103** | **0.16** | **0.23** | 0.297 | 0.366 |
| **ResNet** | 0.0211 | 0.0614 | 0.129 | 0.194 | 0.247 | 0.291 | **0.327** |

Table 3: Navier Stokes equations with $R = 100000$ : Test average MSE per time-step for a Jointly Trained system and one with a pre-trained estimator fine-tuned on the Navier-Stokes data, for different forecasting horizons.

In this section, we study the performances on test samples for the different methods described with Euler and Navier-Stokes dynamics. Pre-training (PT) is always done with Euler dynamics, even for the Navier-Stokes experiments.

Table 1 shows results for Euler dynamics. We can observe that :

- JT SSE clearly diverges. This is an overfitting problem that we have not been able to solve. SSE acts as an encoder-decoder architecture: all relevant information about the past observations is encoded into a single initial state, and the forecasting operator then generates a whole sequence of future state estimations. This seems to be a hard task without the additional prior given by the structure of the real physical state (which we have in the PT pre-trained setting).

- All other three introduced methods along with the ResNet perform relatively well, even for long training horizons. Our algorithms beat the baselines, even if it is only by a small margin for the ResNet.

- As expected, the PT SSE performs well for short to medium forecasting horizons but loses acccuracy in the long run. This is due to the fact that all information is supposed to be stored and rolled forward in a Markovian fashion while the MSRE algorithms, as well as the ResNet and ConvLSTM baselines, work in an auto-regressive fashion over the observations which makes it easier to use information from the past.

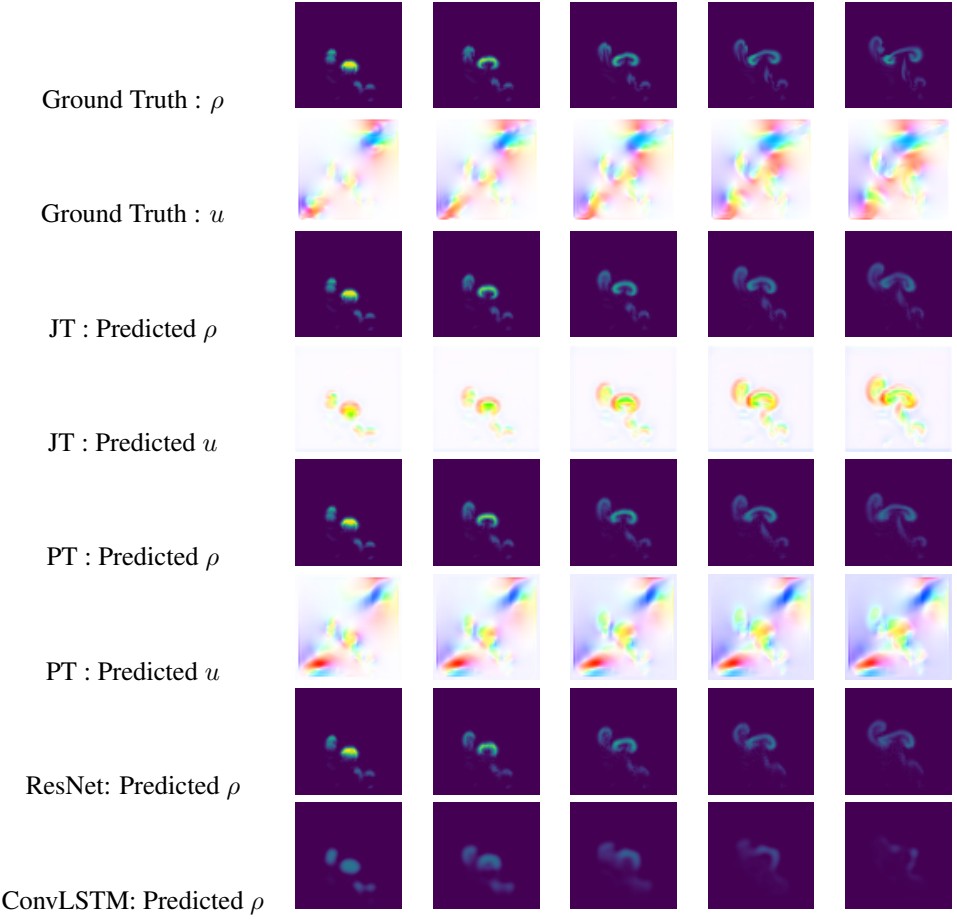

Figure 2: Predicted dynamics for the Euler equations for horizons $t + l = 4, 8, 12, 16, 20$, 1 horizon per column. All algorithms have received as input the same 3 initial images (not shown here). Top 2 rows, correspond to the simulation which in our context is the groundtruth. The first row is the generated density and the second one is the velocity vector field. Rows 3 and 4 give the predictions obtained with a JT MSRE model, rows 5 and 6 the ones for the PT SSE one. Bottom rows correspond to the Resnet and the ConvLSTM. Note that, for the ResNet, the state is the concatenation of observations, and for the ConvLSTM, the states are not interpretable. In order to visualize the velocity $u$, we use the color code used in Baker et al. (2011)
.

- We have justified in section 2 why the ResNet should perform well for this forecasting problem. We argue further in appendix C about the concatenation of observations used as input for the ResNet being a sufficient state representation. This explains the good results obtained by this very standard network.

- Seen from this perspective, compared to the ResNet, JT MSRE only adds a learned transformation of the state optimized for forecasting. This can explain why this network performs so well.

- Figure 5 shows that all three models manage to learn boundary conditions, which are kept constant throughout all datasets.

- Let us also note that the 30 steps we are considering here is somewhat unreasonable : The studied equations are chaotic in nature and in the long run even the simulation which is taken as ground truth here diverges. Thus, the most meaningful results are those for earlier horizons.

Figure 2 shows prediction examples from the different models and largely confirms the observations above. Let us also notice the difference between the SSE and the JT state representations : While the first presents a structure which is very close to the true physical state, the second has a representation which cannot be interpreted as a velocity field but is rather close to the predicted observation itself.

For the remaining part of the paper, we will focus on the PT SSE and the JT MSRE networks which seem to be the most interesting according to the observations above.

Tables 2 and 3 consider the Navier-Stokes equations. As before, we use two algorithms: a jointly trained JT MSRE and a pre-trained estimator PT SSE, in this case pre-trained using Euler equations : Euler dynamics is thus used as structural prior information for the more complex Navier Stokes dynamics. Note that this is clearly an imperfect prior, especially when the viscosity is high (*i.e.* when $R$ is low). We then have a choice between keeping this imperfect estimator, hoping that the forecasting operator will make up for its shortcomings, or fine-tuning it on observations of Navier Stokes dynamics. We try the two options, the fine-tuning being done by jointly training the whole system, starting with a pre-trained estimator, while dividing the learning rate of the estimator's optimizer algorithm by a factor of 100. Results show that fine-tuning yields clearly better results. Moreover, we can also see that the JT algorithm still gives very good results and that the ResNet is consistently beaten by our methods.

## 5.3 Forecasting performance with smaller datasets

|  | $t_0 + 1$ | $t_0 + 5$ | $t_0 + 10$ | $t_0 + 15$ | $t_0 + 20$ | $t_0 + 25$ | $t_0 + 30$ |
|---|---|---|---|---|---|---|---|
| Jointly trained | **0.0015** | 0.0111 | 0.0311 | 0.056 | **0.083** | **0.10** | **0.14** |
| Fine-tuned Pre-Trained estimator | 0.0020 | **0.010** | **0.020** | **0.042** | 0.096 | 0.18 | 0.28 |
| ResNet | 0.0090 | 0.028 | 0.067 | 0.114 | 0.163 | 0.210 | 0.255 |

Table 4: Navier Stokes equations with $R = 5000$ keeping only **half the initial dataset** for training : Test average MSE per time-step for a Jointly Trained system, a system with an estimator **pre-trained on Euler dynamics** fine-tuned on the Navier-Stokes data and a ResNet, for different forecasting horizons.

|  | $t_0 + 1$ | $t_0 + 5$ | $t_0 + 10$ | $t_0 + 15$ | $t_0 + 20$ | $t_0 + 25$ | $t_0 + 30$ |
|---|---|---|---|---|---|---|---|
| Jointly trained | **0.0091** | **0.047** | **0.10** | 0.18 | 0.24 | **0.29** | **0.33** |
| Fine-tuned Pre-Trained estimator | 0.011 | 0.049 | **0.10** | **0.16** | **0.23** | 0.30 | 0.37 |
| ResNet | 0.0290 | 0.0843 | 0.168 | 0.241 | 0.299 | 0.348 | 0.389 |

Table 5: Navier Stokes equations with $R = 100000$ keeping only **half the initial dataset** for training : Test average MSE per time-step for a Jointly Trained system, a system with an estimator **pre-trained on Euler dynamics** fine-tuned on the Navier-Stokes data and a ResNet, for different forecasting horizons.

|  | $t_0 + 1$ | $t_0 + 5$ | $t_0 + 10$ | $t_0 + 15$ | $t_0 + 20$ | $t_0 + 25$ | $t_0 + 30$ |
|---|---|---|---|---|---|---|---|
| Jointly trained | 0.012 | 0.114 | 0.304 | 0.458 | 0.568 | 0.645 | 0.703 |
| Fine-tuned Pre-Trained estimator | **0.0021** | **0.0131** | **0.0323** | **0.0595** | **0.0972** | **0.1452** | **0.2008** |
| ResNet | 0.011 | 0.037 | 0.085 | 0.139 | 0.193 | 0.244 | 0.293 |

Table 6: Navier Stokes equations with $R = 5000$ keeping only **one fourth of the initial dataset** for training : Test average MSE per time-step for a Jointly Trained system, a system with an estimator **pre-trained on Euler dynamics** fine-tuned on the Navier-Stokes data and a ResNet, for different forecasting horizons.

In order to further explore the difference in generalization performance between the different algorithms which we found to be efficient, we trained them on smaller datasets. Tables 4 and 5 show test results with halved datasets. We can see that all three algorithms are quite robust to this diminution of available training data which only slightly affects performance.

However, when we make data even scarcer, in some cases as shown in table 6 where $R = 5000$, the pre-trained network performs more consistently. This is understandable as those dynamics are quite complex (here the fluid is very viscous), the JT having more unconstrained parameters this explains its inability to generalize in this case while the ResNet does better but still sees its performance very

affected by the lack of training of data. We are convinced that this robustness of pre-trained networks is a very important feature and should be studied further as it could prove useful in many real-world settings, especially if combined with more unsupervised algorithms such as JT.

## 6  RELATED WORK

The dynamical system view on NN is not new. In the 90s, several papers developed the analogy between recurrent neural networks and dynamical systems. To quote only a few references Pearl-mutter (1995) offers a nice review of state of the art on RNNs and their training algorithms including convergence and stability ; Tsoi & Back (1994) review RNN architectures with local recurrent connections together with their interpretation as non linear IIR/FIR filters ; Haykin (2009) introduces RNNs with regard to state estimation in dynamical systems. More recently, the success of ResNet-like architectures has motivated new developments and interpretations of NNs from a dynamical systems perspective. The interpretation of residual blocks as implementing a simple forward Euler numerical scheme for ODE is highlighted by several authors such as Chang et al. (2018b); Lu et al. (2018); E (2017). For example, Lu et al. (2018) shows how different ResNet inspired modules can be considered as specific numerical schemes (forward or backward Euler, Runge Kutta, etc..). Chang et al. (2018b) links the depth of ResNet architectures with the size of the discretization step used in ODE. In Gomez et al. (2017); Chang et al. (2018a), the connection with the numerical integration schemes is pushed further by exploiting the idea of reversibility in order to build architectures for which all activations can be reconstructed from the next layer, thus alleviating the need for storing the states of a network and allowing to build larger architecture . Ruthotto & Haber (2018) develop PDE inspired ResNet like models and establish stability results. This line of work concerns networks for static data and targets the development of alternative architectures or training algorithms.

Several works have already attempted to learn PDEs from data (Crutchfield & Mcnamara (1987), Alvarez et al. (2013)). More recently, Rudy et al. (2017a) use sparse regression on a dictionary of differential terms to recover the underlying PDE. In Raissi & Karniadakis (2018), Raissi et al. (2017), they propose recovering the coefficients of the differential terms by deriving a GP kernel from a linearized form of the PDE. For dynamic data, Kim et al. (2018) proposes NN models for fluid simulations for computer graphics applications. The objective is to accelerate fluid simulation w.r.t. classical solvers. They learn to generate 2D and 3D fluid velocities (corresponding to states in our examples), using a fully supervised approach. This share some technical similarities with our own work but the approach and objectives are clearly different. A fully supervised NN implementing a multistep numerical scheme is also used in Long et al. (2018) for learning PDEs from simulations. Raissi (2018) develops a NN framework for learning PDEs from data. In this work either the form of the PDE or the variable dependency is supposed to be known and the approach is again fully supervised over the state.

## 7  CONCLUSION

In this work, we address the problem of forecasting spatio-temporal dynamics governed by differential equations, where information on the system state comes from incomplete observations.

We have presented a general framework separating the task in two problems, state estimation and forecasting, and proposed several instances of this framework in order to accommodate the weakly supervised estimation step. This approach is tested on families of equations governing the dynamics of fluids which are known to be challenging. We have explored empirically some of the properties of the algorithms and evaluated their performance and behavior on high resolution simulated data. We have been able to show that imposing physical priors over the estimator, by pre-training it on simple dynamics then fine-tuning on the target equations, can be beneficial. This is especially true when data is scarce. Jointly training the system without any supervision gives surprisingly good results, especially when lots of data are available.

There are many ideas to explore in order to extend the results. Exploring which physical constraints could be beneficial to the training and to the discovery of physically meaningful models is an important issue. Moreover, we still face difficulties when trying to capture long-term velocity dynamics in the pre-trained setting and this has to be improved if we want to achieve a better performance for the PT SSE model. Our approach also has to be tested on other families of differential equations.

Furthermore, while we have been only considering simulated data, we must obviously aim for data from real-world settings.

As a final note, while we have only considered our physically inspired models for physics motivated datasets, we think that the insights provided by the different settings empirically studied in this work might prove useful for other tasks regarding the forecasting of dynamical systems.

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

## A NAVIER-STOKES AND EULER EQUATIONS

Let us consider a fluid blob of surface $\delta S$, velocity $u$ and density $\rho$. This blob has a pressure $p$ which exerts a force $-\nabla p \delta S$ and is in a gravitational field $g$ so that Newton's second law gives :

$$\rho \delta S \frac{Du}{Dt} = (-\nabla p + \rho g)\delta S$$

where $\dfrac{Du}{Dt}$ expresses the acceleration of the blob thus being the temporal derivative of the function $t \to u(x(t), y(t))$ which can also be expressed as :

$$\frac{Du}{Dt} = \frac{\partial u}{\partial t} + (u \cdot \nabla)u$$

Moreover, we will also make the incompressibility hypothesis which can be mathematically translated to :

$$\nabla \cdot u = 0$$

meaning that $u$ is of null divergence.

Finally, by definition, the mass of the blob we are studying is conserved so that :

$$\frac{D\rho}{Dt} = \frac{\partial \rho}{\partial t} + (u \cdot \nabla)\rho = 0$$

Those equations are the so called *Euler equations* and form the system :

$$\begin{cases} \dfrac{\partial u}{\partial t} + (u \cdot \nabla)u = -\dfrac{\nabla p}{\rho} + g \\[2mm] \dfrac{\partial \rho}{\partial t} + (u \cdot \nabla)\rho = 0 \\[2mm] \nabla \cdot u = 0 \end{cases}$$

When deriving the Euler equations, we never considered the particular nature of the fluid. This is taken into account by considering an additional force in the previous derivation : viscosity. Even for small values, this additional force is of great importance : For example, while air usually has quite small viscosity, without it one wouldn't be able to explain why planes fly.

For Newtonian fluids[2], viscosity can be taken into account with an additional term $\nu \nabla^2 u$ where $\nu = \dfrac{\mu}{\rho}$ is the *kinematic viscosity*. This gives us the following equation system :

$$\begin{cases} \dfrac{\partial u}{\partial t} + (u \cdot \nabla)u = -\dfrac{\nabla p}{\rho} + g + \nu \nabla^2 u \\[2mm] \dfrac{\partial \rho}{\partial t} + (u \cdot \nabla)\rho = 0 \\[2mm] \nabla \cdot u = 0 \end{cases}$$

Those are the *Navier-Stokes equations* which are central to fluid dynamics. We can easily see how the Euler equations can be considered as a simplification of this system, found by taking $\nu = 0$.

The importance of viscosity in a fluid, and thus a way to assess its qualitative behaviour, cannot be measured through $\nu$ alone as it has to be compared against the importance of advection. The most commonly used quantity to describe it is the *Reynolds number*. If we take $U$ to be a typical speed and $L$ a typical distance of the studied flow then it can be expressed as :

$$R = \frac{UL}{\nu}$$

To understand its meaning, let us recall that we have $(u \cdot \nabla)u \sim \dfrac{U^2}{L}$ and $\nu \nabla^2 u \sim \dfrac{\nu U}{L^2}$ so that :

$$R \sim \frac{(u \cdot \nabla)u}{\nu \nabla^2 u}$$

---

[2]Some fluids can behave quite differently and this is translated into a different viscosity term

which means that $R$ is the ratio of the inertia and the viscosity terms. Thus, if $R$ is small, it means that the fluid is very viscous, thus behaving more like honey ; If $R$ is large, the viscosity term is small and the fluid behaves more like water (which doesn't always mean that every such fluid can be modelled with the Eulerian approximation as viscosity can play a role even when the corresponding term is small).

## B  LERAY PROJECTION : EULER AND NAVIER STOKES EXPRESSED IN THE FORM OF EQUATION 1

The two systems equation A and equation A are not of the form equation 1 as we still have the pressure variable $p$ as well as the null divergence constraint. Let us show how we can transform those equations in order to get rid of it.

The Helmholz-Leray decomposition result states that for any vector field $a$, there exists $b$ and $c$ such that :

$$a = \nabla b + c$$

and

$$\nabla \cdot c = 0$$

where $\nabla \cdot$ is the divergence operator. Moreover, this pair is unique up to an additive constant for $b$. Thus, we can define a linear operator $\mathbb{P}$ by :

$$\mathbb{P}(a) = c$$

This operator is a continuous linear projector which is the identity for divergence-free vector fields and vanishes for those deriving from a potential.

Let us take a solution of NS and apply $\mathbb{P}$ on the first equation of equation A, we have, as $u$ is divergence free from the third equation and as $g$ derives from a potential :

$$\frac{\partial u}{\partial t} = -\mathbb{P}[(u \cdot \nabla)u] + \nu \mathbb{P}(\nabla^2 u)$$

where permuting derivation and $\mathbb{P}$ is justified by the continuity of the operator[3].

Thus, if $u$ is solution to equation A, it is also a solution to :

$$\begin{cases} \dfrac{\partial u}{\partial t} = -\mathbb{P}[(u \cdot \nabla)u] + \nu \mathbb{P}(\nabla^2 u) \\ \\ \dfrac{\partial \rho}{\partial t} = -(u \cdot \nabla)\rho \end{cases}$$

which is of the form of equation 1

Conversely, the solution of the above system is such that :

$$u_t = \int \frac{\partial u}{\partial t} = \int -\mathbb{P}[(u \cdot \nabla)u] + \nu \mathbb{P}(\nabla^2 u)$$

which gives, by exchanging $\mathbb{P}$ and the integral[4] :

$$u_t = \mathbb{P}[\int -(u \cdot \nabla)u + \nu \nabla^2 u]$$

so that $u$ is automatically of null divergence by definition of $\mathbb{P}$. The two systems are thus equivalent.

---

[3]One can use a finite difference approximation to show it for example.

[4]To prove this, we just have to take a sum approximation to the integral and use again the linearity then the continuity of $\mathbb{P}$.

## C    EXPRESSING THE STATE AS A FUNCTION OF OBSERVATIONS

All our work relies on the assumption stated in equation equation 3 which supposes that states can be expressed as a deterministic function of observations. This is obviously not always true : If the observations are uncorrelated with the true state, one can certainly not retrieve it. Thus, if one considers a physical system modelled by a state $X_t$, then we are actually estimating the conditional expectation of this state given a sequence of observations $(Y_{t-k+1}, ..., Y_t)$. Here, the stochasticity could come from some noise in the operator $\mathcal{H}$ as well as from the fact that the observations may not contain enough information to reconstruct the state. In other words, our assumption amounts to suppose that there exists $k$ such that $X_t$ is $\{Y_{t-k}, ..., Y_t\}$-measurable.

Another take on this matter would be to consider that what we want is simply to construct a state which is sufficient for the forecasting. In other words, in the language of dynamical systems, we want an embedding of the attractor our system is in which can be constructed with a sequence of observations. The celebrated Takens embedding theorem tells us precisely that, for a prevalent class of observation functions $\mathcal{H}$, there is a $k$ such that, by putting $X_t = (Y_{t-k+1}, ..., Y_t)$, there exists $F$ for which $F(X_t) = X_{t+1}$. In our case, this means that, as long as all we want is to have a state allowing the forecasting of observations, which is the underlying motivation of the JT models, our assumption is verified for more or less any observation operator $\mathcal{H}$[5].

All this being said, our overall intuition is that this assumption is not very restrictive in the deterministic case and it should work as long the observations are meaningful enough. In the noisy case, which is not studied in this paper, one would have to take into account the sensibility of the studied dynamics to initial conditions which, in the case of chaotic systems, necessitates to control well enough the conditional variance at the estimation step.

## D    COMPARISON OF TRAINING SEQUENCE LENGTHS

|          | $t_0 + 1$ | $t_0 + 5$ | $t_0 + 10$ | $t_0 + 15$ | $t_0 + 20$ | $t_0 + 25$ | $t_0 + 30$ |
|----------|-----------|-----------|------------|------------|------------|------------|------------|
| $l = 2$  | **0.004** | 0.0478    | 0.195      | 0.434      | 0.763      | 1.20       | 1.79       |
| $l = 5$  | 0.006     | **0.038** | **0.105**  | **0.175**  | 0.239      | 0.2956     | 0.345      |
| $l = 8$  | 0.007     | 0.044     | 0.10       | 0.176      | **0.235**  | **0.284**  | **0.327**  |
| $l = 11$ | 0.01      | 0.052     | 0.103      | 0.189      | 0.245      | 0.292      | 0.332      |

Table 7: Forecasting with Jointly Trained model JT MSRE on Euler simulations: Test average MSE per time-step for different training sequence lengths $l$, for different forecasting horizons. Testing horizon $l = 8$.

|          | $t_0 + 1$ | $t_0 + 5$ | $t_0 + 10$ | $t_0 + 15$ | $t_0 + 20$ | $t_0 + 25$ | $t_0 + 30$ |
|----------|-----------|-----------|------------|------------|------------|------------|------------|
| $l = 2$  | 0.03      | 0.165     | 0.340      | 0.5        | 0.645      | 0.78       | 0.912      |
| $l = 5$  | **0.01**  | **0.054** | 0.104      | 0.203      | 0.287      | 0.369      | 0.446      |
| $l = 8$  | 0.011     | 0.059     | **0.102**  | 0.179      | 0.234      | 0.288      | 0.341      |
| $l = 11$ | 0.012     | 0.064     | 0.107      | **0.178**  | **0.223**  | **0.267**  | **0.313**  |

Table 8: Forecasting with pre-trained Estimator on Euler simulations: Test average MSE per time-step for different training sequence lengths $l$, for different forecasting horizons. Testing horizon $l = 8$.

Tables 7 and 8 show results for different training horizons $l$ for the two algorithms PT SSE and JT MSRE. We can clearly see that, at least up to a certain point, the longer the training horizon, the better the long term performance for both algorithms. This is to be expected : A longer training horizon acts as an additional constraint for the forecasting operator and thus ensures a better generalization.

This observation is especially true for the PT algorithm : The improvements observed for the PT algorithm in table 8 are higher than the improvements for the JT algorithm in table 7 and the PT algorithm supersedes JT at longer horizons. This seems to indicate that the state representation with

---

[5]The precise statement is actually that the operators who work forms a dense subset.

| | $t_0 + 1$ | $t_0 + 5$ | $t_0 + 10$ | $t_0 + 15$ | $t_0 + 20$ | $t_0 + 25$ | $t_0 + 30$ |
|---|---|---|---|---|---|---|---|
| **Scheduled Sampling** | **0.007** | 0.044 | 0.100 | 0.176 | **0.235** | **0.284** | **0.327** |
| **No Teacher Forcing** | 0.008 | **0.043** | **0.108** | **0.175** | 0.236 | 0.289 | 0.335 |
| **Systematic Teacher Forcing** | 0.026 | 0.215 | 0.108 | 0.567 | 0.663 | 0.736 | 0.795 |

Table 9: Forecasting with Jointly Trained system : Test average MSE per time-step with Scheduled Sampling, Systematic Teacher Forcing and No Teacher Forcing, for different forecasting horizons.

| | $t_0 + 1$ | $t_0 + 5$ | $t_0 + 10$ | $t_0 + 15$ | $t_0 + 20$ | $t_0 + 25$ | $t_0 + 30$ |
|---|---|---|---|---|---|---|---|
| **Scheduled Sampling** | 0.011 | 0.06 | **0.102** | **0.179** | **0.234** | **0.288** | **0.341** |
| **No Teacher Forcing** | 0.011 | **0.058** | **0.102** | 0.182 | 0.24 | 0.294 | 0.346 |
| **Systematic Teacher Forcing** | **0.009** | 0.076 | 0.225 | 0.374 | 0.5 | 0.605 | 0.694 |

Table 10: Forecasting with pre-trained estimator : Test average MSE per time-step with Scheduled Sampling, Systematic Teacher Forcing and No Teacher Forcing, for different forecasting horizons.

a physical meaning is more stable for long term forecasting and benefits more from this additional constraining. However, we still keep $l = 8$ for the other experiments for computational reasons.

## E    COMPARISON OF SAMPLING STRATEGIES

Tables 9 and 10 compare the 3 sampling strategies introduced in section 3, namely systematic teacher forcing, no teacher forcing and scheduled sampling. For the latter, we consider a decreasing sequence $(\epsilon_i)_i$, starting from 1 and converging to 0, so that, at time $i$, we pick $Y_{t-1}$ with probability $\epsilon_i$ and $\widehat{Y}_{t-1}$ with probability $1 - \epsilon_i$. This amounts to starting with one-step predictions then gradually increasing the time horizon as the model improves. The $(\epsilon_i)_i$ sequence has to decay at the same pace as the error of the model and thus has to be chosen carefully depending on the trained model. In this work, we have considered the exponential scheme :

$$\epsilon_i = \alpha^i$$

Results show that using systematic teacher forcing performs poorly, which is to be expected as it is equivalent to a training horizon of 1, and, surprisingly, not using any teacher forcing works remarkably well, almost as well as scheduled sampling while training much faster which might be used to accelerate training. However, in all the experiments, we used scheduled sampling nonetheless.

## F    ADDITIONAL FIGURES

We provide below some example of forecasting images obtained for Navier Stokes equations. The setting of the experiments is introduced in the main text in section 5.

Ground Truth : $\rho$

Ground Truth : $u$

JT : Predicted $\rho$

JT : Predicted $u$

PT : Predicted $\rho$

PT : Predicted $u$

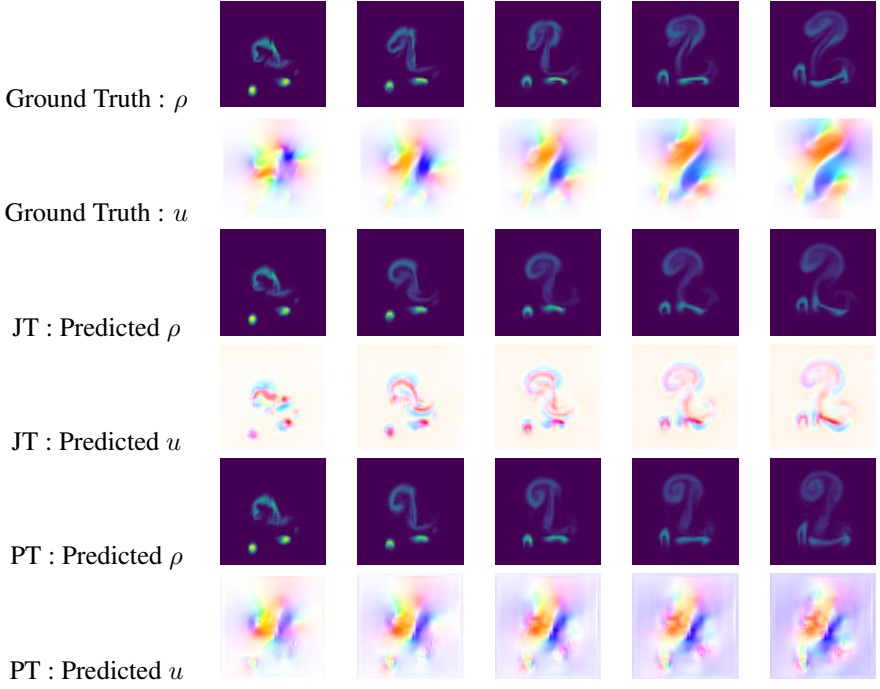

Figure 3: Predicted dynamics for the Navier Stokes equations with $R = 100000$ for $t + l = 4, 8, 12, 16, 20$, each algorithm having received the same 3 initial images (not shown here). Top 2 rows correspond to the simulation which in our context is the ground truth. First row is the generated density and second one is the velocity vector field. For the latter, each point is two dimensional, the color code is provided in the appendix. Rows 3 and 4 give the predictions obtained with a JT MSRE model, rows 5 and 6 the ones for the PT one.

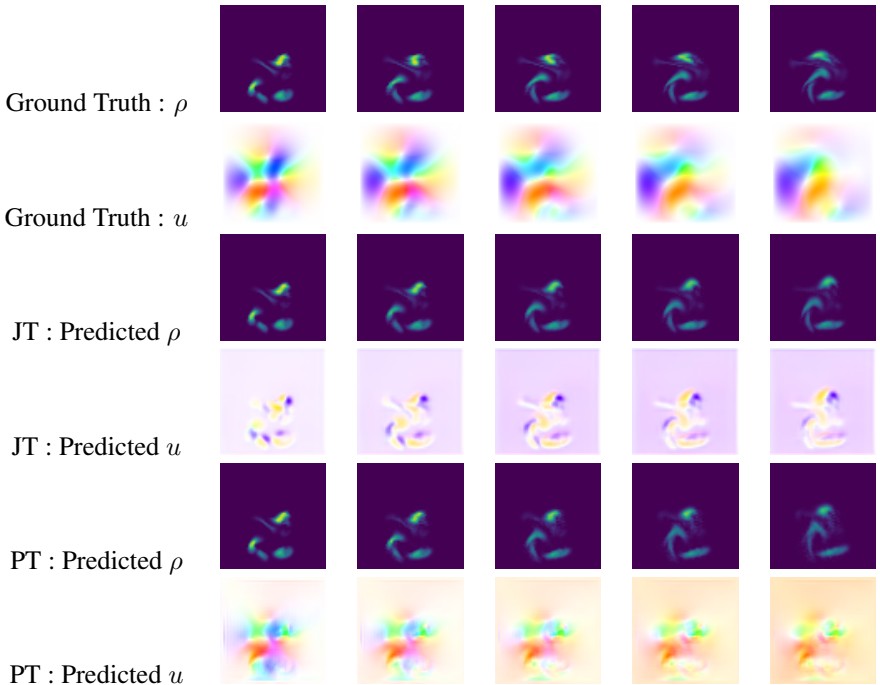

Figure 4: Predicted dynamics for the Navier Stokes equations with $R = 5000$ for $t + l = 4, 8, 12, 16, 20$, each algorithm having received the same 3 initial images (not shown here). Top 2 rows correspond to the simulation which in our context is the groundtruth. First row is the generated density and second one is the velocity vector field. Rows 3 and 4 give the predictions obtained with a JT MSRE model, rows 5 and 6 the ones for the PT one.

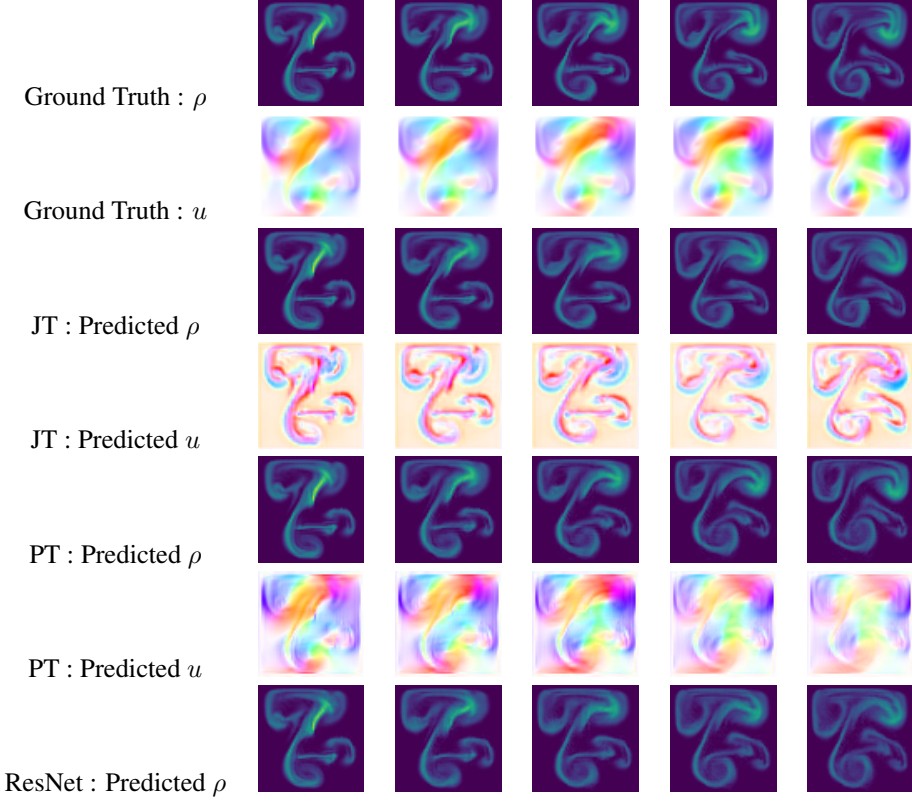

Figure 5: Predicted dynamics for the Navier Stokes equations with $R = 100000$ for $t + l = 4, 8, 12, 16, 20$, each algorithm having received the same 3 initial images (not shown here). Top 2 rows, correspond to the simulation which in our context is the groundtruth. First row is the generated density and second one is the velocity vector field. For the latter, each point is two dimensional, the color code is provided in the appendix. Rows 3 and 4 give the predictions obtained with a JT MSRE model, rows 5 and 6 the ones for the PT SSE one while the last row is for the ResNet. Here we can see the boundary conditions coming into effect : All three algorithms manage to learn them.

