# OpenReview forum: "Learning Partially Observed PDE Dynamics with Neural Networks"
_ICLR.cc/2019/Conference_

### Official Review · AnonReviewer3 · 2018-11-05
**Some questions in lieu of a review, for now**

**Rating:** 5
**Confidence:** 3

**Review:**

I feel like I am missing something about this paper, so rather than a review, this is just mainly a long question making sure I understand things properly.  Ignore the score for now, I'll change once I get a clearer picture of what's happening here.

The network you propose in this paper is motivated by solving PDEs where, as in (1), the actual solution as they are computed numerically depends on the current spatial field of the state, as well as difference operators over this field (e.g., both the gradients and the Laplacian terms).  So, I naturally was assuming that you'd be designing a network that actually represented state as a spatial field, and used these difference operators in computing the next state.  But instead, it seems like you reverted to the notion of "because difference operators can be expressed as convolutions, we use a convolutional network", and I don't really see anything specific to PDEs thereafter, just general statements about state-space models.

Am I understanding this correctly?  Why not just actually use the PDE-based terms in the dynamics model of an architecture?  Why bother with a generic ResNet? (And I presume you're using a fully convolutional ResNet here?)  Wouldn't the former work much better, and be a significantly more interesting contribution that just applying a ResNet and a generic U-Net as a state estimator?  I'm not understanding why the current proposed architecture (assuming I understand it correctly) could be seen as "PDE guided" in all but the loosest possible sense.  Can you correct me if I'm misunderstanding some element here?

---

> ### Author Response · Authors · 2018-11-26
> **Answer to Reviewer3 (part 1/2)**
>
> We appreciate very much your approach, we are sorry that the paper has not been sufficient to clearly explain our overall approach. We will try to make up for this in this answer and clarify the revised version of the paper in accordance.
>
> First of all, we agree that the title might have been misleading. Taking into account the reviews, we have decided to change it to “Learning partially observable PDEs with neural networks” which states more explicitly what is done. In other words, what we are trying to do is to forecast space time processes which are driven by unknown PDEs having access only to partial state measurements.
>
> The direction that you are suggesting which consists, as we understand it, in designing a specific architecture with explicit differential terms which might appear in the studied PDEs has actually been investigated by several recent papers. There has been different approaches, the main ones being :
>
> -Those which consist in numerically calculating (with finite difference schemes) candidate differential terms then regressing against measured data. Here complete states are supposed to be available and the goal is mainly to retrieve the form of the underlying PDE. Schaeffer’s “Learning partial differential equations via data discovery and sparse optimization” or Rudy et al. ‘s “Data-driven discovery of partial differential equations” are good examples of this approach. In Raissi et al ‘s “Physics Informed Deep Learning”, a similar view is taken but with automatic differentiation instead of explicit numerical schemes, still constructing a dictionary of differential terms.
>
> -A more hybrid approach was the one followed by Long et al ‘s “PDE-NET” paper where convolution filters are constrained to approximate differential operators of a certain order but still have learnable parameters. Again, the main goal here is to find the terms of the underlying equation with the complete states supposed to be available.
>
> The work above is indeed very interesting and promising. We are actually convinced that many of their ideas will be relevant to our future research.
>
> However, in this paper, we take a different point of view :
>
> -First of all, our goal is not to recover the underlying equation but rather to find efficient methods for forecasting. In this regard, we have found that standard non-constrained ResNets, when supervised and evaluated with complete states, are very powerful at forecasting without overfitting training data. Thus, additional explicit constraints in the forecasting operator didn’t seem necessary, especially as we don’t want to make any general hypothesis regarding the differential terms which might intervene in the underlying unknown PDE or the way those terms can be computed. Moreover, from early experiments, those constraints didn’t lead to improvements in predictive performance, while adding numerical instabilities and sometimes difficulties in training.
>
> -The most important point is that we place ourselves in the more realistic setting of having access only to partially observable states of the underlying PDEs, where most of the variables cannot be directly measured. One has then to estimate a state making the forecast possible and our goal is to see whether it is still possible to build a system which consistently succeeds in doing so, with few or no prior information over the dynamics governing the studied data. Ideally, we would like the approach to work for any dynamical system.
>
> In order to solve this problem :
>
> -We present a general and flexible framework where forecasting is decomposed into two steps, which closely follows the way applied physicists work with this kind of PDEs translated into a NN architecture.
>
> -While keeping the models very generic, as you rightfully pointed out, we study different variants of algorithms obtained through this architecture (SSE and MSRE), with different levels of prior injection into the estimator (pretrained and joint training) and apply this to an important class of PDEs, with promising results up to a relatively long horizon.
>
> -Our results show that the unsupervised version works surprisingly well while pretraining with a simplified model (which we view as injecting a structural prior in the state), here Euler as a prior to Navier-Stokes, is interesting when data is scarce. There is more empirical evidence for this last observation in the revised version of the paper.

---

> > ### Author Response · Authors · 2018-11-26
> > **Answer to Reviewer3 (part 2/2)**
> >
> > Obviously, it would be very interesting to merge the two approaches. For instance, one aspect we will research in the coming times is to study the interplay between constraining the forecasting operator f, for example by adding constraints to the filters or by replacing them with ODE solvers and see how this affects the estimator. This can be seen as injecting prior information into f and it is an important question to see when to do this given a certain level of knowledge about the dynamics, how to do it efficiently and if additional knowledge improves the forecasting accuracy.
> >
> > On the other hand, it is important to note that we have showed how our method actually gives us a way to easily inject physical priors into the estimator e in the partially observable setting. There is also the question of imposing more principled structural constraints in its architecture but we are convinced that this depends on the studied problem and thus one would have to study different classes of PDEs to look for appropriate NN models. This is also a direction of research which is interesting for us.
> >
> > We hope those few points clarify our endeavors and we have tried to improve our presentation in the revised version of the paper so that this can be understood more clearly. We have also added more experiments analyzing the performance of the models at different levels of data scarcity showing how injecting prior knowledge can improve forecasting.

---

### Official Review · AnonReviewer2 · 2018-11-05
**A good step in learning Navier-Stokes equations, but lack compelling results**

**Rating:** 5
**Confidence:** 5

**Review:**

+ An interesting idea to learn the hidden state evolution and the state-observation mapping jointly
+ The experiments on Euler's equation are slightly better than ResNet for 30 steps ahead forecasting in terms of MSE
+ The paper is clearly written and well-explained

- The model is not new: ResNet for state evolution and  Conv-Deconv for state-observation mapping
- The difference between ResNet and the proposed framework is not significant, ResNet is even better in Figure 2
- Missing an important experiment:  test whether the model can generalize, that is to forecast on different initial conditions than the training dataset
- How does the model compare with GANs (Y. Xie* , E. Franz* and M. Chu* and N. Thuereyy, “tempoGAN: A Temporally Coherent, Volumetric GAN for Super-resolution Fluid Flow”)?

---

> ### Author Response · Authors · 2018-11-26
> **Answer to Reviewer2**
>
> Thank you very much for your review and comments. Let us address some of your concerns.
>
> In this work, we present a general formulation for forecasting dynamical systems using neural networks, in a setting where we do not fully observe its state. The aim of our work is to conceive a framework that is applicable to a wide range of dynamical systems, not to focus solely on the problem of fluid dynamics prediction which was merely an example taken as application. For this reason, we have used standard neural network architectures (Resnet and UNet), that may not be novel but have proven to work well for a large range of different tasks. Nonetheless, conceiving task-specific architectures and integrating them into our formulation is possible, and indeed an interesting research direction. We wanted this work to focus on the generic framework applied to the partially observable setting and not on any problem specific issues.
>
> Regarding the generalization of our models, in our datasets, in the training, the validation as well as in the test sets, all sequences are generated randomly and independently, meaning the location of densities and the intensity and direction of initial fluid flows are sampled at random and independently for each sequence (there are 200 of those in each test set which were produced after the training of the models). The only parameters that are fixed through all experiments are the boundary conditions, for which generalization could be interesting to study but this was not the scope of this work and for which we have added an additional figure (in the additional figures section of the appendix) showing how models learn them, and, of course, the dynamics within each dataset. In other words, in the test phase (MSE results and figures), our system successfully forecasts starting with initial conditions which it has never seen during training.
>
> As for the comparison with the TempoGAN paper, while it is indeed a very interesting work in the area of physics-aware neural networks, we do not see it as relevant for what we propose in this work as the authors solve a different task : while having access to the complete state of the system (including the velocity flow and vorticity which they find to have a regularizing effect) at all times, their goal is to solve the super-resolution problem where a coherent high-resolution flow is obtained from a low-resolution dynamic whereas we solve the forecasting problem at a fixed resolution with access to only a projection of the complete state of past times. However, it is an interesting direction of research to see whether it is possible to improve forecasting results by using generative networks, for example in the estimation step.  At this stage, it is still not very clear how this could be implemented but we think it is worth exploring.
>
> Your remarks regarding the very good performance of ResNets are indeed an important point. This standard architecture can actually be seen as an instance of our framework, by simply considering the sequence of k observations as the system’s state, and setting e to be equal to Id, the identity operator. Those good performances show that the generic ResNet architecture is a particularly well suited one for dynamical systems : Actually, when we started experimenting in the fully observable situation (H=Id), just using this architecture allowed obtaining near perfect test results (we will include those experiments in the revised version of the paper). However, while this naive state representation seems sufficient for forecasting, and there are indeed classical theoretical arguments for this to be true, it is not necessarily the most efficient one : With the other proposed architectures, we wish to find alternative state representations better suited for forecasting where structural priors on the state can be enforced. As systems grow in complexity and data is scarcer, a good representation becomes more important. This can be seen for example with the comparative advantage of the PT model as compared to JT and the ResNet when diminishing the dataset size in the NS experiments we have added in the revised version of the paper. And, while we have only showed this on a single example, one has to keep in mind the fact that equations in real-world systems are much more complex with data in scarcer quantities, as it is costlier to obtain, so that this kind of prior can prove useful in many situations.

---

### Official Review · AnonReviewer1 · 2018-11-05
**Review of "Learning space time dynamics with PDE guided neural networks"**

**Rating:** 6
**Confidence:** 3

**Review:**

I very much like the aim of this work. This is a problem of interest to a wide community, which as far as I'm aware hasn't yet had much focus from the deep learning community. However, perhaps in part because of this, the paper reads as naive in places. Pages 1-4 are all background saying nothing new, but ignoring the effort made on this problem by other communities. There has been some work done o this problem within statistics, and within the Gaussian process community, to which no reference is made at all by the paper.

There are two novelties as far as I can see (these may or may not be novel - but they were novel to me). The first is use of NNs to model the system. The second is the multiple state restimation (MRSE) on page 5. I struggled to get a feeling about how successful these two aspects of the work are. The results section is difficult to follow, and doesn't compare the method to existing methods and so there is no baseline to say that this is successful or not. Thus I find it hard to judge the execution of the idea. What I really want to know reading a paper like this is should I use this approach? Because there is no comparison to existing methods, it leaves me unsure.

Other comments:
- Is the title correct? I don't see how these are PDE guided NNs? You've used data from a PDE to train the network and as a test problem. A PDE guided NN would, for me, know something about the dynamics (compare with recently work in the GP community where kernels are derived that lead to GPs that analytically obey simple PDEs).
- There is an obvious link to work in the uncertainty quantification community, particularly around the use of multi-fidelity/ multi-level simulation. This paper is likely to be of interest to them and the link could be more explicit.
- Page 3, after eq 2 - there is notation used here that is undefined Y_{t-k}^t
- The simplifying assumption on page 3 is very strong and unlikely to hold for many systems. But it isn't clear to me whether this is necessary or not? Presumably if it doesn't hold then we may still get an approximation that could be useful, but it is just that we lose any guarantee the method will work.
- I thought the MSRE idea was interesting. It wasn't very well explained or motivated, and it was unclear to me whether it works well or not from the results, or whether it is novel to this paper or not. But I'd like to have read more about it.
- Is the trick in Section 8.2 original to this paper? If so, it seems a nice idea (I've not checked the detail).
- Most of section 8.1 strikes me as unnecessary.
- There are quite a few typos. In particular, words such as Markovian, Newtonian should be capitalised.

---

> ### Author Response · Authors · 2018-11-26
> **Answer to Reviewer1 (part 1/2)**
>
> Thank you very much for your extensive and detailed comments. We will try to address your remarks and concerns in this answer. We will also try to make clearer some points that we might have gone through too quickly in the paper.
>
> You are right to mention the GP community, they are very active and pioneering in the area of learning dynamical systems governed by PDEs. We have added more references to the works we have knowledge of to the revised version of our paper (such as [1, 2, 3]). Please feel free to mention any paper that you think would be relevant to add. However, it seems for us that these methods, although promising and useful in many cases, cannot directly be applied to our problem :
>
> -These methods have access to knowledge about the underlying PDE. Typically, in [1, 2, 4],  a dictionary of differential terms intervening in the PDE is supposed to be known. In [3], the PDE is known up to a linear forcing term. In [1], in the non-linear setting, only a few unknown parameters are learned. We place ourselves in a more prior-agnostic context, where we don’t make such a constraining hypothesis.
>
> -These methods all rely on numerical schemes to discretize the PDE. Specifically, in [3], they use backward Euler method for time discretization for all recovered PDEs. In [4], they add a polynomial interpolation to smooth the discrete numerical scheme.  For long-term forecasting, designing and selecting these numerical schemes should be highly dependant on the underlying PDE, using generic discretization may lead to large forecast errors and even to numerical instability. Our formulation does not suffer from this problem : selecting the appropriate discretization scheme is directly incorporated in the learning problem and ResNets have proven to be quite robust.
>
> -Another very important difference, and it is the central issue which is addressed in this work, is the fact that we do not have access to complete states, only to partial state observations. This setting is very common when tackling real world problems in applied physics but we have no knowledge of other approaches which tackle it in the machine learning community. In particular, we don’t know of such in the GP community and it is thus difficult to compare our work with their results.
>
> Precisely for the reason that you mentioned regarding the subject being a new one in the deep learning community, we have struggled to find a strong baseline that we can compare our models to. We have chosen the ConvLSTM model, which is now a classical one in statistical forecasting, and a standard standalone ResNet, which can be seen as an instance of our framework and proved to be a strong baseline. All of the few other works on using deep learning to solve differential equations that we have knowledge of assume the state is fully observable, which is not the case in our setting. On the other hand, data assimilation algorithms used by physicists assume the equations are known and use them to estimate the true state while our system implicitly learns them through training data. Ultimately, one would want to test our learned models against those algorithms in real-world settings where no explicit exact equations are known to show that they work better than the hand-crafted approximations currently used. We will be working in this direction and this paper is a first step paving the way towards this objective.
>
> Thus, what we have tried to build is a framework which allows to perform this task in two different ways : one which is classical in dynamical systems (SSE) and the second one (MSRE) which seemed natural to derive from our framework (so natural that there are certainly variants similar to it available in the time series forecasting literature but we couldn’t find a precise reference or work using it). Intuitively, while the SSE is constrained to compress all relevant information for any time horizon into the estimated state, which is difficult, the MSRE works in a fully auto-regressive way and goes back to the observation space at each time-step which we feel should help, especially when the number of time-steps is greater.

---

> > ### Author Response · Authors · 2018-11-26
> > **Answer to Reviewer1 (part 2/2)**
> >
> > We have also explored another direction in this paper, regarding the injection of structural prior through pretraining : this gives us the pretrained (PT) and jointly-trained (JT) alternatives. This allows us to explore whether we are able to constrain the structure of the estimated state with a simplified model for example, which can be a way to use prior knowledge on the governing PDE like its general form.
> >
> > Thus, our framework is PDE-guided in the sense of its general construction separating clearly estimation from forecasting, in the use of the ResNet architecture, which arguably implements learned finite difference schemes, used as forecasting operator and, more importantly, in the PT case where there is indeed some knowledge of the complete state structure injected through pretraining. However, we do agree with you about the fact that the title might be a little misleading as we never explicitly input any analytical equation into our system. Thus, we have decided to modify the title to make it more explicit. This is precisely our goal : building a generic system, more or less constrained depending on the available knowledge, which is able to learn dynamics through measured observations only.
> >
> > We present results for all those four different alternatives, weighing the strengths and weaknesses of each and trying to explain them intuitively. We will try to make this part of the presentation more palatable and clearer in the revised version as it was obviously confusing for many reviewers.
> >
> > Regarding the simplifying assumption of page 3, it is actually a subtle question to know whether the measure function is enough to reconstruct a state which allows to consistently make the forecast. There are at least two different points of view. One is the probabilistic one, where the estimated state is seen as the conditional expectation of the state given the k observations. The second one comes from the theory of dynamical systems and uses Takens’s embedding theorem which proves that a state can be reconstructed for a dense class of observation functions, as long as k is big enough. We didn’t want to complexify the presentation but we will add a paragraph in the appendix expanding on this question : our general opinion is that, in most cases, as long as the observations give a meaningful signal, there should be a minimal value of k which works. It is even more difficult to know how an error in estimating the state would propagate to the resulting forecast. This would depend on the chaoticity of the studied system and how sensitive it is to initial conditions so it should be considered very carefully on a case by case basis.
> >
> > For other remarks : The section 8.1 is for readers who don’t have prior knowledge of fluid dynamics so that it helps build some intuition of the studied equations. The projection trick in section 8.2 is actually classical in computational fluid dynamics but we had never seen it applied in the deep learning community so we felt it might be interesting to mention it in the paper. We also agree with the link you make between our work and that of the multi-fidelity community but we are still unsure about how such a link could be implemented, it is one of the interesting future research directions we want to pursue. Again, if you have in mind specific papers which might be relevant to our work, we welcome any suggestion.
> >
> > Finally, thank you for pointing those typos, we will do our best in correcting them for the revised version of this paper.
> >
> > [1]: Hidden physics models: Machine learning of nonlinear partial differential equations, https://www.sciencedirect.com/science/article/pii/S0021999117309014
> > [2]: Linear Latent Force Models using Gaussian Processes, https://arxiv.org/abs/1107.2699
> > [3]: Machine learning of linear differential equations using Gaussian processes, https://www.sciencedirect.com/science/article/pii/S0021999117305582
> > [4] Data-driven discovery of partial differential equations”, Rudy et al., http://advances.sciencemag.org/content/3/4/e1602614

---

### Author Response · Authors · 2018-11-26
**Modifications in the revised version of the paper**

In the revised version we have now uploaded, we have tried to take into account the different remarks and concerns of reviewers. This included:
-Revising the second part of the introduction to describe more clearly our contributions;
-Restructuring the experiments section which was confusing for many reviewers;
-Adding standard ResNet experiments for all datasets, as it is indeed a strong model in our setting;
-Adding experiments with even smaller datasets to study generalization;
-Completing the related works sections with some of the references suggested by the reviewers;
-Adding a section to the appendix expanding on the simplifying assumption stated in section 3;
-Adding a figure with an additional sample from the test set showing how boundary conditions are dealt with by the different models;
-Changing the title to make it more explicit;
-Correcting differents typos in the paper.

We have also added clarifications and corrected mistakes throughout the text.

---

### Meta-Review · Area_Chair1 · 2018-12-10
**Interesting area but doesn't meet quality or clarity standards**

**Confidence:** 4
**Recommendation:** Reject

**Metareview:**

This paper introduces a few training methods to fit the dynamics of a PDE based on observations.

Quality:  Not great.  The authors seem unaware of much related work both in the numerics and deep learning communities.  The experiments aren't very illuminating, and the connections between the different methods are never clearly and explicitly laid out in one place.
Clarity:  Poor.  The intro is long and rambly, and the main contributions aren't clearly motivated.  A lot of time is spent mentioning things that could be done, without saying when this would be important or useful to do.  An algorithm box or two would be a big improvement over the many long english explanations of the methods, and the diagrams with cycles in them.
Originality:  Not great.  There has been a lot of work on fitting dynamics models using NNs, and also attempting to optimize PDE solvers, which is hardly engaged with.
Significance:  This work fails to make its own significance clear, by not exploring or explaining the scope and limitations of their proposed approach, or comparing against more baselines from the large set of related literature.